# Classification of Clinical Outcomes in Hospitalized Asian Elephants Using Machine Learning and Survival Analysis: A Retrospective Study (2019–2024)

**DOI:** 10.3390/vetsci12100998

**Published:** 2025-10-16

**Authors:** Worapong Kosaruk, Veerasak Punyapornwithaya, Pichamon Ueangpaiboon, Taweepoke Angkawanish

**Affiliations:** 1Faculty of Veterinary Medicine, Chiang Mai University, Chiang Mai 50100, Thailand; veerasak.p@cmu.ac.th; 2Elephant, Wildlife, and Companion Animals Research Group, Chiang Mai University, Chiang Mai 50200, Thailand; 3Research Center for Veterinary Biosciences and Veterinary Public Health, Chiang Mai University, Chiang Mai 50100, Thailand; 4National Elephant Institute, Elephant Hospital, Thai Elephant Conservation Center, Forest Industry Organization, Ministry of Natural Resources and Environment, Lampang 52190, Thailand; pichamoneye@gmail.com (P.U.); taweepoke@gmail.com (T.A.)

**Keywords:** Asian elephant, treatment, classification model, machine learning, survival analysis, clinical outcome

## Abstract

**Simple Summary:**

This study analyzed five years of clinical records from 467 Asian elephants admitted to Thailand’s largest referral hospital. Using four routinely collected variables: age, sex, disease group, and length of hospital stay, we developed a machine learning model to classify treatment outcomes. A Random Forest algorithm delivered the best performance, suggesting potential for distinguishing clinical outcomes. Elephants with acute illnesses such as herpesvirus-hemorrhagic disease or toxin exposure typically deteriorated quickly, while those with dental or renal conditions required longer treatment. Such classification models may be useful for in-hospital monitoring and decision support, helping veterinarians anticipate clinical trajectories, communicate prognosis more clearly, and guide care decisions.

**Abstract:**

Captive Asian elephants (*Elephas maximus*) frequently present to hospitals with complex, multisystemic diseases, yet veterinarians lack objective tools to predict and classify clinical outcomes. Decision-making often relies on experience or anecdote, and few studies have applied data-driven approaches in wildlife medicine. This study developed a machine learning–based classification model using routinely collected clinical data. A total of 467 medical records from hospitalized elephants at Thailand’s National Elephant Institute (2019–2024) were retrospectively analyzed. Four variables (age, sex, disease group, and length of stay [LOS]) were used to train four classification algorithms: Random Forest, eXtreme Gradient Boosting, Naïve Bayes, and multinomial logistic regression. The Random Forest model achieved the highest classification performance (accuracy = 86.3%; log-loss = 0.374), with disease group, LOS, and age as key predictors. Survival analysis revealed distinct hospitalization trajectories across disease groups: acute conditions like elephant endotheliotropic herpesvirus-hemorrhagic disease and toxicosis showed rapid early declines, whereas dental and renal cases followed more prolonged courses. Our findings demonstrate the preliminary feasibility of outcome classification in elephant care and highlight the potential of clinical data science to improve in-hospital prognostication, monitoring, and treatment planning in zoological and wildlife medicine.

## 1. Introduction

The Asian elephant (*Elephas maximus*) is an endangered species experiencing continued population decline across its natural range. In Thailand, an estimated 3200 individuals live in the wild, with over 3800 in captivity [1,2]. Captive elephants frequently require veterinary intervention for a wide range of conditions including trauma, gastrointestinal disease, systemic infections, nutritional imbalances, and age-related conditions [3,4]. In response, specialized hospitals have emerged to meet this growing demand. Among these, the National Elephant Institute (NEI) hospital in Lampang Province serves as Thailand’s primary referral center for elephant medicine, offering both inpatient care and mobile veterinary outreach.

The NEI hospital treats a diverse caseload, ranging from minor injuries to severe multisystemic illnesses, and receives elephants from private owners, conservation programs, and government agencies. Although the majority of admissions involve captive elephants, the NEI hospital also occasionally treats wild elephants. Between 2019 and 2024, approximately ten wild-born elephants were admitted, primarily orphaned or injured calves. Across all hospitalized elephants, prognoses vary widely: elephant endotheliotropic herpesvirus-hemorrhagic disease (EEHV or EEHV-HD), for example, is highly lethal in juveniles despite aggressive treatment performed [5], while older elephants with chronic musculoskeletal or dental issues may recover after prolonged hospitalization [4]. This variability in outcomes complicates in-hospital decision-making, especially in the absence of structured classification frameworks. At present, outcome assessment in elephant hospitals relies largely on clinical judgment, which may be inconsistent across cases.

Machine learning and artificial intelligence (AI) are increasingly being applied in veterinary medicine to improve diagnostic and classification capabilities. This trend mirrors their transformative impact in human healthcare, offering enhanced data-driven analytics and decision support in animal care [6,7]. Currently, structured classification models are now widely adopted in human critical care and veterinary medicine, supporting clinical monitoring, resource prioritization, prognosis, and family/owner communication [8,9,10,11,12]. Recent studies demonstrate the utility of machine learning in classifying outcomes for equine colic cases. In one study, AI algorithms accurately identified horses needing surgery and those likely to survive an acute colic episode [13]. Even in wildlife health contexts, outcome classification has demonstrated utility. Recent reviews emphasize that triage protocols should be evidence-based in wildlife rehabilitation centers to maximize survival and welfare [14]. For instance, risk-based triage models for stranded marine mammals have improved care allocation and reduced prolonged suffering in patients with low survival likelihood [15]. These precedents emphasize the potential value of classification analytics in veterinary wildlife care, particularly where treatment resources are limited and timely decision-making is ethically critical.

This study aimed to address that gap by developing a machine learning–based classification model for hospitalized elephants using routinely collected clinical data. We retrospectively analyzed five years of records from the NEI hospital and trained four machine learning models (Random Forest, eXtreme Gradient Boosting [XGBoost], Naïve Bayes, and multinomial logistic regression) based on four admission-accessible features: age, sex, disease group, and length of hospital stay (LOS). Additionally, survival analysis was conducted to explore time-to-event patterns across disease categories and enhance interpretability of model predictions. By converting routine hospital records into a classification framework, this study introduces one of the first formal classification models in elephant medicine. The findings could support the growing role of clinical data science in wildlife care systems and align with One Health principles by bridging conservation practice, veterinary decision support, and the use of artificial intelligence in veterinary medicine [16,17].

## 2. Materials and Methods

### 2.1. Data Collection and Variables

We retrospectively reviewed clinical records of all elephants admitted to the NEI hospital in Lampang Province, Thailand (GPS: 18.4024, 99.3319), between January 2019 and December 2024 (N = 470). Data were extracted from standardized paper-based forms and included age (in years), sex (male/female), primary disease group, and admission and discharge dates. After excluding three cases with missing dates and one without disease classification, 467 cases were retained (Figure 1).

Length of hospital stay (LOS) was calculated as the number of days from admission to discharge or censoring (Figure 2), and was treated as a dynamic proxy for clinical trajectory—not as a retrospective outcome. LOS was computed up to the censoring date to preserve causal precedence and prevent data leakage. Elephants admitted prior to 1 January 2019 but still hospitalized during the study window were retained. Their LOS was calculated from the original admission date to censoring or discharge, which explains durations exceeding the 2019–2024 window.

Disease classification was based on clinical diagnosis and presenting complaints. Three elephant veterinarians independently grouped each case into one of ten groups: integumentary, gastrointestinal, eye, musculoskeletal, EEHV, reproductive, toxicity, dental/oral, urinary/renal, and miscellaneous. Cases with unclear localization (e.g., systemic inflammation) were grouped as “Miscellaneous.” Rare diagnoses (<5 cases), including tetanus, adrenal insufficiency, and respiratory condition, were collapsed into an “Other” category to mitigate sparsity-related overfitting and preserve model stability. A full breakdown of diagnoses within the “Other” group is provided in Appendix A.

### 2.2. Descriptive Analysis

Demographic and clinical characteristics were summarized using standard descriptive statistics. Continuous variables (e.g., age, LOS) were reported as mean ± standard error (SE); categorical variables (e.g., sex, disease group, and outcome) as frequencies and percentages. Cross-tabulations were used to visualize outcome distributions across disease groups.

### 2.3. Model Development

Four supervised classification models were trained to classify in-hospital treatment outcomes (deceased, ongoing, recovered) using age, sex, disease group, and LOS. The models included: 1) Random Forest, implemented via the ranger engine with 500 trees [18]; 2) XGBoost, with 500 boosting rounds and a learning rate of 0.05 [19]; 3) Naïve Bayes, assuming conditional feature independence [20]; and 4) Multinomial logistic regression, as a linear baseline [21]. These models were selected to represent a range of linear and non-linear classifiers, balancing interpretability and predictive power.

Categorical variables were dummy-coded; near-zero variance features were excluded. Continuous variables were left unscaled due to the robustness of tree-based models to monotonic transformations.

The final dataset (N = 467) was randomly split into 80% training (N = 373) and 20% held-out test set (N = 94) using stratified sampling by outcome. Five-fold cross-validation was used to assess performance based on multiclass accuracy and log-loss. To address class imbalance (48 deceased, 23 ongoing, 396 recovered), inverse-frequency class weights were computed using the training set only and applied during model training using the case_weights function in ranger. No threshold adjustments or tuning procedures were performed on the test set.

The final locked model was trained on the weighted training set and evaluated once on the held-out test set. Metrics included confusion matrices, per-class precision, recall, F1 score, and overall multiclass ROC AUC. Confidence intervals were calculated via bootstrapping. Model calibration was assessed using Brier scores and decile-based reliability plots. A separate binary classifier (recovered vs. deceased) was also trained to focus on terminal outcomes.

Additionally, we implemented a simplified “Day-0” model using only age, sex, and disease group as predictors, excluding LOS. Model structure, preprocessing steps, and evaluation followed the same framework as the main model. This approach aimed to simulate early-phase monitoring utility.

### 2.4. Survival Analysis

Kaplan–Meier analysis was used to estimate time to clinical resolution, defined as either hospital discharge or death. The event was treated as a composite outcome, with elephants still hospitalized at data cut-off considered censored. Survival curves were stratified by disease group, initialized at 1.0, and plotted with 95% confidence intervals. Median survival time was calculated per group. In groups with low event rates and long admissions (e.g., dental/oral or urinary/renal), upper bounds were right-censored.

Cox proportional hazards regression was used to evaluate associations between covariates and time-to-event outcomes. The hazard function was defined as:$$h(t)=h_{0}(t)\times \exp\left(\beta_{1}X_{1}+\beta_{2}X_{2}+\cdots +\beta_{p}X_{p}\right)$$
where $h\left(t\right)$ is the hazard at time t, $h_{0}\left(t\right)$ is the baseline hazard, and $\beta_{1}X_{1}+\beta_{2}X_{2}+\cdots +\beta_{p}X_{p}$ represents covariate effects. Covariates included age, sex, and disease group. Proportional hazards assumptions were assessed using Schoenfeld residuals (cox.zph function) (Appendix A). Minor violations were noted for the disease group variable and interpreted cautiously. While survival analysis was conducted separately from the machine learning framework, its outputs were used to contextualize risk trajectories and validate the time dependency of model classification.

To further clarify the dynamics of recovery versus mortality, we additionally performed cause-specific Cox regression, modeling time to death and time to recovery as competing events. In each model, alternative outcomes were treated as censored.

### 2.5. Software and Reproducibility

All analyses were performed in R (v4.5.1). Core packages included tidymodels 1.3.0, ranger 0.17.0, xgboost 1.7.11.1, naivebayes 1.0.0, survival 3.8–3, survminer 0.5.0, and forestmodel 0.6.2. Analytical R codes are provided in Appendix A.

## 3. Results

### 3.1. Descriptive Findings

The mean age of hospitalized elephants was 33.7 ± 1.0 years (females: 34.2 ± 1.3 years; males 32.9 ± 1.6). Full demographic and clinical characteristics are presented in Table 1.

Proportion of treatment outcomes by disease group is presented in Figure 3. Presenting conditions most commonly involved the integumentary, gastrointestinal, and eye systems. Most elephants recovered, while fewer were classified as deceased or ongoing at the time of data censoring. Outcome distributions varied substantially by disease category. Mortality was highest in elephants diagnosed with EEHV, toxicity, or rare conditions grouped under “Other”. In contrast, integumentary and eye diseases were strongly associated with recovery. Notably, some high-mortality categories (e.g., toxicity) demonstrated broad outcome variability, suggesting heterogeneous clinical trajectories within the same disease group.

### 3.2. Classification Model Performance

Model comparison from five-fold cross-validation is summarized in Table 2. Among the four classifiers tested, the Random Forest model achieved the best overall performance form the test dataset (accuracy: 86.3%, log-loss: 0.374), outperforming XGBoost, Naïve Bayes, and logistic regression. Although Naïve Bayes produced comparable accuracy, it exhibited poorer probability calibration as reflected by higher log-loss.

Evaluation on a held-out test set (N = 94) demonstrated reasonable discrimination by the baseline Random Forest (macro-weighted ROC AUC: 0.81; class-wise recall: 0.50, F1: 0.54). Average precision across classes was 0.67. However, performance was lower for deceased and ongoing outcomes, consistent with limited sample representation for these classes. To address this, we further implemented an inverse-frequency weighted Random Forest model. This approach improved performance for minority classes: macro-average recall rose to 0.74 and macro F1 score to 0.64 (Appendix A).

Variable importance was assessed from the weighted Random Forest model using impurity-based metrics. Disease-related predictors were grouped to improve interpretability (Figure 4), while the complete list of variable contributions is presented in Appendix A.

For a secondary binary classifier (deceased vs. recovered), this model yielded 74% accuracy with a ROC AUC of 0.87 (Appendix A).

The Day-0 model achieved an overall accuracy of 80% and a multiclass ROC AUC of 0.71. While performance for the recovered class remained high (F1 = 0.89), both deceased and ongoing outcomes showed poor precision and recall (Appendix A).

Model calibration metrics further supported reliability. For the multiclass model, reliability plots showed strong agreement for the deceased class, moderate calibration for recovered, and variable performance for ongoing class, the latter likely reflecting the small sample size (N = 23) (Appendix A). The macro-averaged Brier score was 0.082 for multiclass. The corresponding confusion matrix for the final weighted model is shown in Appendix A. Binary model calibration was also acceptable (Brier score = 0.116), with observed and predicted probabilities closely aligned (Appendix A).

To facilitate practical application, we developed a clinician-facing prototype tool comprising: (1) an Excel-based input sheet; (2) the final pre-trained model (final_rf_weighted_model.rds); and (3) user instructions These allow real-time estimation of outcome probabilities based on four input variables (Age, Sex, Disease group, LOS). These files are provided in Appendix A as a downloadable ZIP.

### 3.3. Time-to-Event Outcomes

Length of hospital stay varied widely across disease group (Table 1), with median survival times ranging from 2 days (EEHV) to over 60 days in dental/oral and urinary/renal conditions (Table 3). Acute presentations (e.g., EEHV, toxicity, gastrointestinal disease) were associated with higher early-phase mortality and shorter hospital stays. In contrast, musculoskeletal, reproductive, and dental/oral cases often required prolonged treatment and were associated with more favorable outcomes.

Kaplan–Meier survival curves illustrated distinct divergence between acute and chronic disease trajectories (Figure 5). EEHV and toxicity displayed steep declines within the first week, while integumentary, reproductive, and musculoskeletal disorders exhibited more gradual reductions in survival probability. Full risk tables for all group are provided in Appendix A.

Cox proportional hazards regressions confirmed that age and disease group were significant predictors of time to discharge or death (Figure 6). Gastrointestinal disease (Hazard Ratio [HR] = 1.92), EEHV (HR = 2.47), and toxicity (HR = 2.14) were associated with significantly increased hazards, suggesting more rapid clinical deterioration and increased urgency for intervention.

In the cause-specific Cox regression analyses, EEHV (HR = 87.1, 95% CI: 20.5–371.1), toxicity (HR = 23.3, 95% CI: 4.4–124.0), and “Other” disease group (HR = 24.6, 95% CI: 5.4–112.1) showed the highest hazards for death (Appendix A). Conversely, reproductive (HR = 0.35) and urinary/renal diseases (HR = 0.28) were strongly associated with delayed recovery (Appendix A). Cause-specific survival curves confirmed rapid deterioration in acute disease groups and longer recovery times in chronic cases (Appendix A).

## 4. Discussion

This study represents the initial effort to apply machine learning to classify clinical outcomes in hospitalized Asian elephants. Drawing from five years of records at Thailand’s National Elephant Institute hospital, our findings suggest that a Random Forest classifier, using four routinely recorded variables (age, sex, disease group, and length of hospital stay [LOS]), can provide a preliminary framework for stratifying clinical outcomes (deceased, ongoing, recovered). This aligns with broader trends in veterinary data science and highlights the potential role of AI-driven tools in supporting in-hospital clinical decision-making in wildlife settings.

Our results align with growing literature on machine learning applications in veterinary medicine. Random Forest models have demonstrated comparable accuracy in predictive models for canine parvovirus, equine colic, and neonatal foal survival, often in the 80–85% range [11,22,23]. For example, a recent equine study found that a tuned Random Forest model outperformed 14 other algorithms, attaining ~85% accuracy in predicting colic surgery outcomes [22]. In our study, the Random Forest outperformed logistic regression, Naïve Bayes, and XGBoost, reaffirming the advantages of non-linear ensemble methods in modeling heterogeneous clinical data. This model had also robustness to heterogeneity and class imbalance, supporting its use for in-hospital monitoring, especially in wildlife datasets where sample sizes are modest and laboratory diagnostics or electronic health records may be unavailable [11].

Among predictors, age emerged as both biologically intuitive and clinically informative. Juvenile and geriatric elephants exhibited greater risk, consistent with prior reports in both elephant medicine and broader veterinary literature [5,10]. Younger elephants, especially those with EEHV, often deteriorated rapidly [24], while older animals tended to suffer prolonged recovery due to degenerative conditions or comorbidities [4]. Although formal interaction terms were not tested due to sample size constraints, observed trends suggest age may compound risk in specific disease categories, warranting further investigation.

Sex was not a strong predictor of clinical outcome, aligning with prior work in equine and bovine critical care once age and disease status were controlled [25,26]. A nationwide EEHV analysis in Thailand also found no effect of sex on mortality [24]. Nonetheless, rare sex-linked risks in elephants such as dystocia in pregnant females may become more apparent in larger datasets [27]. Including sex retains interpretability and ensures adaptability in future iterations.

Disease group was the most powerful predictor of treatment outcome, comparable in several previous reports in humans [28,29] and animals [30,31,32]. Acute, systemic illnesses such as colic, EEHV-HD, and toxicosis were associated with rapid clinical decline and poor prognosis. These results mirror clinical expectations and veterinary literatures: in domestic horses, approximately 10% of colic cases require surgical emergencies, which delated intervention sharply reducing survival [22], while EEHV-HD in elephants has an overall fatality rate exceeding 70% in some subtypes [24,33,34]. In contrast, integumentary, musculoskeletal, and ocular cases often followed more prolonged but ultimately recoverable courses [35,36]. Our survival analyses, including cause-specific Cox regressions, supported these distinctions: EEHV, toxicity, and gastrointestinal diseases showed markedly elevated hazard of death, whereas reproductive and urinary/renal conditions were associated with delayed recovery, reflecting prolonged but manageable trajectories. The “Other” category, which included rare and complex cases (e.g., tetanus, adrenal insufficiency), also demonstrated poor overall outcomes, highlighting the potential severity of unfamiliar or atypical presentations.

The inclusion of LOS as a model input requires careful interpretation. Although LOS is inherently tied to clinical outcome, we calculated it only up to the censoring date (31 December 2024). Comparable approaches are seen in human Intensive Care Unit (ICU) prediction models, where LOS has been used as dynamic feature reflecting clinical evolution [10]. For instance, Back et al. [37] showed that each additional hospital day increased sepsis risk by 18% within their automated sepsis risk scoring system. Similarly, Jana et al. [38] treated LOS as a joint prediction target alongside critical interventions, illustrating its role as both an outcome and a temporal marker of disease trajectory. As anticipated, model performance declined when restricted to early-phase data. The Day-0 model, which excluded LOS, yielded moderate overall accuracy but failed to discriminate deceased and ongoing outcomes. This underscores the need for temporally informed inputs in effective in-hospital monitoring models. However, LOS input in our study may not fully capture disease chronology in elephants, as many of whom undergo days of camp-based treatment before referral. In conditions such as colic, delayed admission may allow reversible conditions to progress to systemic compromise, echoing patterns observed in horses [39,40]. The absence of data on symptom onset or pre-hospital interventions therefore remains a critical limitation that likely introduces unmeasured variability.

As expected, our model performed best for the recovered class, which dominated the dataset, while the minority classes (deceased and ongoing) were under-represented, leading to lower baseline sensitivity, which comparable to other reports [41,42]. Applying inverse-frequency class weighting improved performance matrixes for these categories, particularly the deceased class. These adjustments are consistent with best practices for imbalanced classification problems, which are known to degrade model reliability if unaddressed [43,44]. For high-stakes decisions, the binary model (recovered vs. deceased, excluding ongoing cases) may offer clearer clinical utility, particularly in urgent triage settings requiring a clear survival prediction. This binary model also achieved good performance (see Appendix A). Nonetheless, we retained the three-class outcomes in the main text to reflect clinical reality: many elephants remain hospitalized with uncertain prognosis at the time of record closure. By explicitly modeling “ongoing” cases, we allow the system to capture real-time uncertainty, while acknowledging that these cases will eventually resolve and require reclassification.

Several limitations must be acknowledged. First, the model was trained on data from a single referral hospital, potentially limiting generalizability. Institutional protocols, clinician preferences, and case mixes vary across settings. External validation is therefore essential and planned in upcoming studies. Second, while our sample size (N = 467) is large by elephant medicine standard, it remains modest in machine learning terms, with limited power to resolve low-frequency diagnoses. Although we addressed this partially by collapsing rare categories into “Other,” larger datasets will enable finer stratification. While this approach stabilized model training, it may reduce interpretability and obscure high-risk conditions. We recommend future studies explore clinically driven grouping schemes to better reflect biological relevance and support targeted interventions. Third, the absence of standardized clinical parameters (e.g., bloodwork, vital signs) likely constrained predictive performance. In canine parvovirus and equine neonatology, integrating basic lab tests has significantly improved classification accuracy [11,26]. Incorporating such data into future models could enhance both sensitivity and clinical interpretability. Fourth, the use of “ongoing” as an outcome may pose conceptual ambiguity, as it reflects a temporary treatment status rather than final result. A survival modeling approach or periodic retraining may be required to accommodate evolving case outcomes over time. Finally, while proportional hazards assumptions were mostly met, minor violations were noted for the disease group variable. These were interpreted cautiously, but may have introduced some bias in time-to-event estimates.

Despite these constraints, our study introduced the practical utility of machine learning in elephant hospital care. The model is lightweight, interpretable, and adaptable to resource-limited contexts. In future iterations, the model could be integrated into mobile dashboards for use by field veterinarians, providing real-time estimates to support in-hospital prognosis monitoring, escalation, and caregiver communication. This work exemplifies the value of clinical data science in wildlife health. Classification tools not only assist individual case management but also contribute to system-level decision-making, improving care efficiency, ethical clarity, and conservation outcomes. As Asian elephants are a keystone species with immense ecological and cultural significance in Thailand, enhancing survival during medical intervention is an urgent priority. Collaboration and data sharing among wildlife facilities will be essential to refine predictive accuracy and ensure that these tools generalize across different settings, ultimately elevating the standard of care for endangered species.

## 5. Conclusions

This study introduced the first machine learning–based classification model for hospitalized Asian elephants, utilizing four routinely available clinical variables to predict clinical outcomes. Despite limitations in sample size and single-center design, the model demonstrated encouraging internal validity and practical interpretability. By converting real-time hospital data into individualized risk estimates, this tool may provide a practical foundation for more real-time monitoring, clinical trajectory stratification, and communication with caregivers. In particular, clinicians working in resource-limited or high-caseload settings may benefit from standardized outcome estimates when deciding on escalation of care or case prioritization. Although preliminary, these findings support future integration of clinical predictive models into routine elephant hospital workflows. Such tools may ultimately enhance treatment efficiency and survival in endangered species, aligning with broader conservation and One Health goals.

## Figures and Tables

**Figure 1 vetsci-12-00998-f001:**
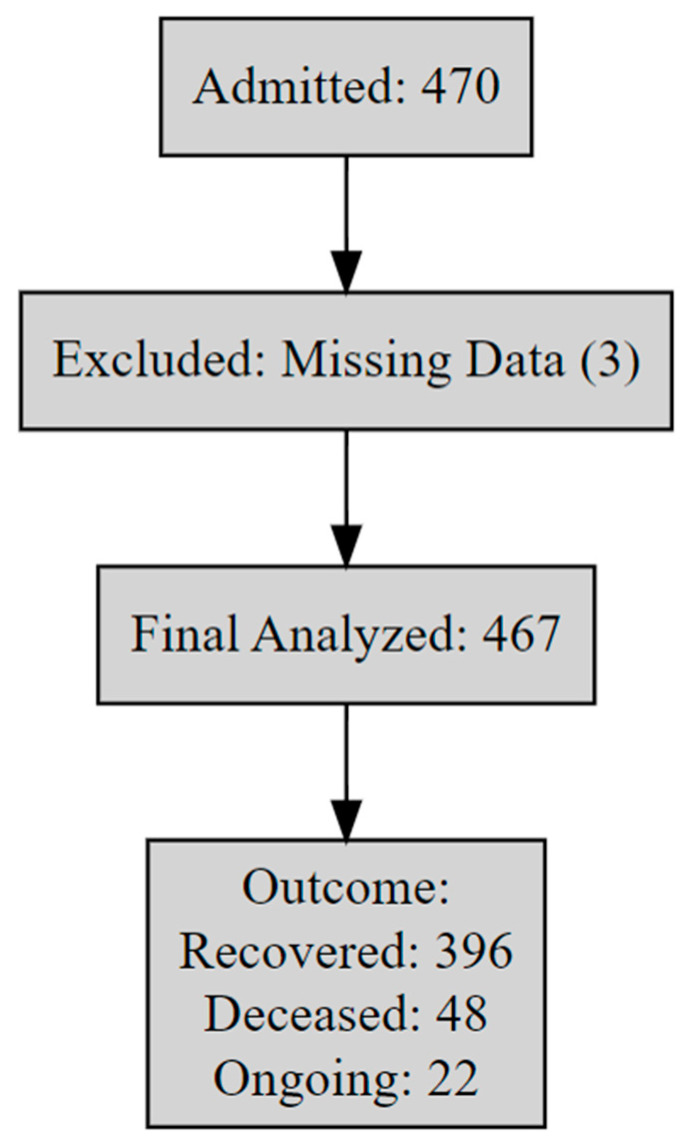
Flow diagram of case inclusion. A total of 470 elephants were admitted between 2019 and 2024; after excluding 3 cases with missing data, 467 were analyzed. Treatment outcomes were classified as of 31 December 2024 into three mutually exclusive categories: (1) Deceased: died during hospitalization; (2) Ongoing: still under care at the censoring date; and (3) Recovered: discharged in stable condition. Although “ongoing” is not a terminal outcome, it was retained to reflect unresolved cases in real-time and allow the model to differentiate them.

**Figure 2 vetsci-12-00998-f002:**
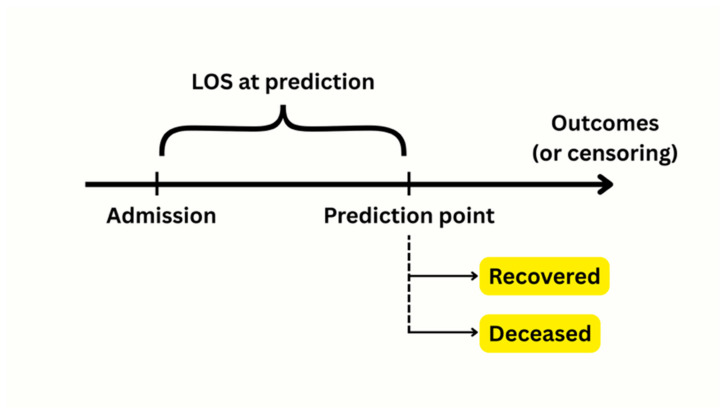
Schematic illustration of the relationship between admission, length of stay (LOS), and outcome in hospitalized elephants. The LOS at classification represents the duration from hospital admission to the point at which the model makes a prognosis. Outcomes (recovered or deceased) occur after this prediction point. Ongoing cases are censored at the cutoff date (31 December 2024). This time-aware structure ensures that LOS precedes the outcome and avoids temporal bias.

**Figure 3 vetsci-12-00998-f003:**
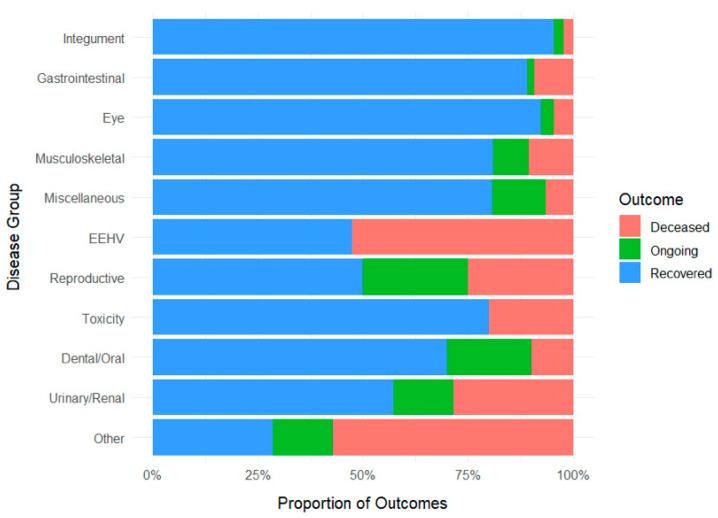
Proportion of treatment outcomes; deceased (red), ongoing (green), and recovered (blue), by disease group. EEHV; Elephant endotheliotropic herpesvirus.

**Figure 4 vetsci-12-00998-f004:**
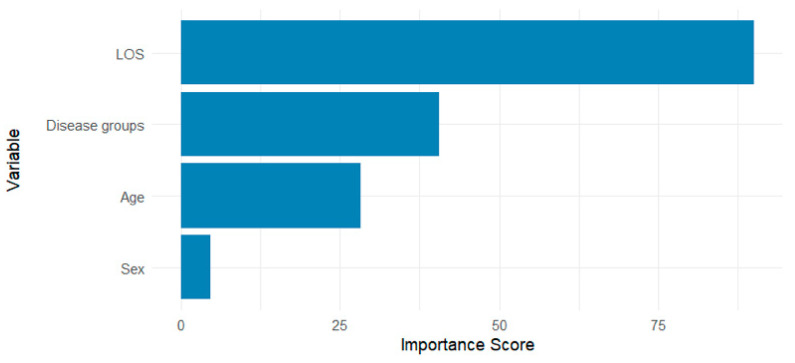
Grouped variable importance from the final weighted Random Forest model classifying hospitalization outcomes in Asian elephants. Disease groups were treated as a single categorical predictor comprising 11 clinical groups. LOS; Length of stay (days).

**Figure 5 vetsci-12-00998-f005:**
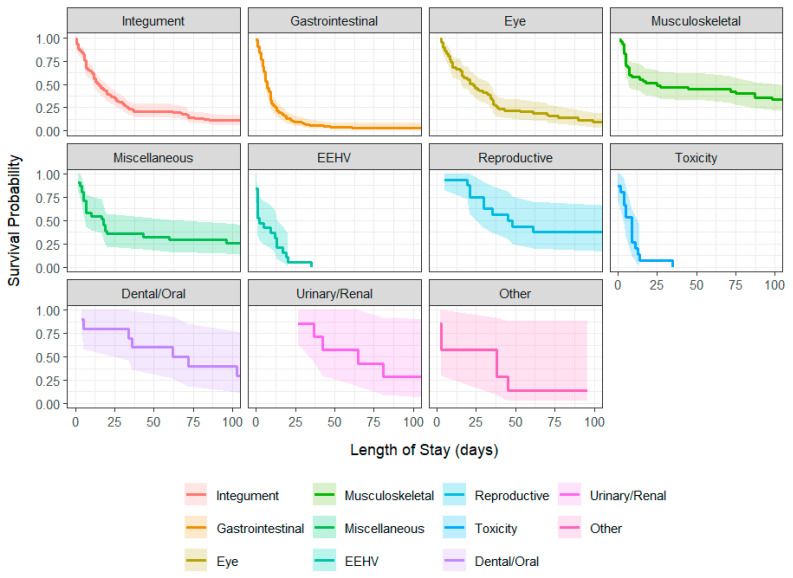
Kaplan–Meier survival curves stratified by disease group, using a composite event of hospital discharge or in-hospital death. Lines represent estimated survival probabilities with 95% confidence ribbons. Horizontal axis shows time since admission (in days). EEHV; Elephant endotheliotropic herpesvirus.

**Figure 6 vetsci-12-00998-f006:**
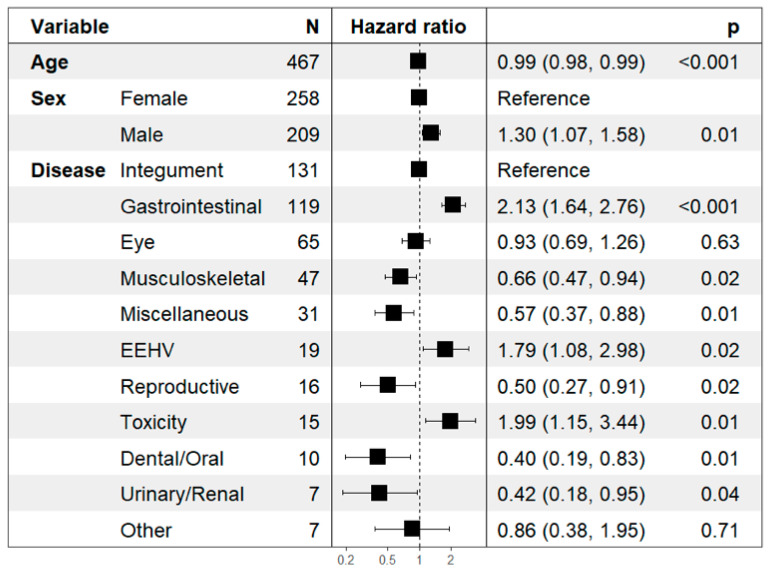
Multivariable Cox regression results for treatment outcome. Hazard ratios with 95% confidence intervals and *p*-values are shown. Age was modeled as a continuous variable; hazard ratios represent the relative risk change per one-year increase in age. Female sex and integument disease group are reference categories. EEHV; Elephant endotheliotropic herpesvirus.

**Table 1 vetsci-12-00998-t001:** Demographic and clinical characteristics of the study cohort (N = 467) by sex and disease group.

Variable	Overall (%)	Deceased (%)	Ongoing (%)	Recovered (%)	Length of Stay (Day) Mean ± SE (Range)
Sex					
▪ Female	258 (55.25%)	27 (10.47%)	13 (5.04%)	218 (84.50%)	102.88 ± 19.38 (0 to 2380)
▪ Male	209 (44.75%)	21 (10.05%)	10 (4.78%)	178 (85.17%)	80.95 ± 17.57 (0 to 1919)
Disease group					
▪ Integument	131 (28.05%)	3 (2.29%)	3 (2.29%)	125 (95.42%)	88.63 ± 23.05 (0 to 1523)
▪ Gastrointestinal	119 (25.48%)	11 (9.24%)	2 (1.68%)	106 (89.08%)	41.24 ± 21.91 (0 to 2380)
▪ Eye	65 (13.92%)	3 (4.62%)	2 (3.08%)	60 (92.31%)	47.72 ± 13.11 (2 to 821)
▪ Musculoskeletal	47 (10.06%)	5 (10.64%)	4 (8.51%)	38 (80.85%)	240.66 ± 76.24 (1 to 1934)
▪ Miscellaneous	31 (6.64%)	2 (6.45%)	4 (12.90%)	25 (80.65%)	201.32 ± 69.10 (2 to 1257)
▪ EEHV	19 (4.07%)	10 (52.63%)	-	9 (47.37%)	7.95 ± 2.21 (0 to 35)
▪ Reproductive	16 (3.43%)	4 (25.0%)	4 (25.0%)	8 (50.0%)	174.19 ± 73.33 (5 to 1058)
▪ Toxicity	15 (3.21%)	3 (20.0%)	-	12 (80.0%)	8.60 ± 2.19 (0 to 35)
▪ Dental/Oral	10 (2.14%)	1 (10.0%)	2 (20.0%)	7 (70.0%)	220.30 ± 110.25 (4 to 1035)
▪ Urinary/Renal	7 (1.50%)	2 (28.57%)	1 (14.29%)	4 (57.14%)	113.14 ± 49.42 (27 to 396)
▪ Other	7 (1.50%)	4 (57.14%)	1 (14.29%)	2 (28.57%)	32.14 ± 12.83 (2 to 96)
Total	467 (100%)	48 (10.30%)	22 (4.72%)	396 (84.98%)	93.06 ± 13.28 (0 to 2380)

EEHV; Elephant endotheliotropic herpesvirus.

**Table 2 vetsci-12-00998-t002:** Cross-validated performance of machine learning models for classifying treatment outcomes in hospitalized elephants. Accuracy and multiclass log-loss were estimated using five-fold cross-validation.

Model	Accuracy (Mean ± SE)	Log-Loss (Mean ± SE)
Random Forest	0.863 ± 0.017	0.374 ± 0.046
XGBoost	0.860 ± 0.021	0.559 ± 0.103
Naïve Bayes	0.849 ± 0.014	2.13 ± 0.217
Logistic	0.858 ± 0.011	0.750 ± 0.125

XGBoost; eXtreme Gradient Boosting.

**Table 3 vetsci-12-00998-t003:** Median survival times (with 95% confidence intervals; CI) by disease group based on Kaplan–Meier estimates. Some upper bounds are undefined due to right-censoring.

Disease	Median (Days)	95% CI Lower	95% CI Upper
Integument (N = 131)	15.0	12	20
Gastrointestinal (N = 119)	7.0	6	9
Eye (N = 65)	23.0	16	34
Musculoskeletal (N = 47)	25	7	111
Miscellaneous (N = 31)	18.0	7	96
EEHV (N = 19)	2.0	1	17
Reproductive (N = 16)	46.5	30	-
Toxicity (N = 15)	9.0	4	13
Dental/Oral (N = 10)	67.0	34	-
Urinary/Renal (N = 7)	65.0	37	-
Other (N = 7)	38.0	3	-

EEHV; Elephant endotheliotropic herpesvirus.

## Data Availability

The raw individual-level clinical records of hospitalized elephants are available from the corresponding author on reasonable request due to privacy and ethical restrictions. All other data, R analytical codes, and the outcome classification tool are provided in the article/Appendix A.

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
