# Peer review of "Classification of Clinical Outcomes in Hospitalized Asian Elephants Using Machine Learning and Survival Analysis: A Retrospective Study (2019–2024)"

_vetsci, 2025, doi:10.3390/vetsci12100998_

Round 1
Reviewer 1 Report
Comments and Suggestions for Authors
The article tackles an interesting area with real clinical value, i.e., to turn routine elephant hospital records into simple outcome predictions. However, in this current form, I advise major revisions. The biggest problem is using the length of stay to predict the final outcome, which essentially involves peeking into the future unless you lock the prediction to a fixed time point. Similarly, the third outcome“ still hospitalized” mixes up with true endpoints. There are also basic consistency issues (e.g., length of stay up to 2,380 days in a 2019–2024 dataset; mismatched case counts) that suggest data or reporting errors. Hyperparameter tuning and software versions should be fully described, and the survival results need complete reporting and PH checks. The manuscript also needs copy-editing and reference cleanup. A clear path to acceptance: redefine the prediction task to avoid leakage, audit and correct the LOS/count issues, strengthen interpretability and labeling reliability, and share de-identified (or synthetic) data with full code.
Reviewer 2 Report
Comments and Suggestions for Authors
The authors have undertaken quite interesting research into the treatment of such a challenging species as elephants. The research conducted and the results presented are very interesting and have both cognitive and practical significance, and consequently, could have a significant impact on the management of sick animals. Therefore, they perfectly fit the scope of species conservation, nature conservation in general, and, above all, saving endangered animals in various disease contexts and their etiologies. In principle, I have no major comments about the manuscript. I only have a few questions and concerns that should be clarified:
- It would be worth adding the phrase "treatment" to the keywords.
- In the introduction, the authors mention the need to treat captive elephants, but are there any instances of attempts to treat wild individuals?
- In my opinion, the "material and methods" section is too long; it should be considered shortening it.
- The first paragraph of the discussion is not a discussion, but rather a conclusion.
- The conclusions are too modest, and there are no clear indications for the potential practical application of these preliminary results to improve the diagnosis and treatment of elephants.
It would be worthwhile to clarify the issues I have described and add them to the paper, which would allow for deeper analysis and perhaps even conclusions. As the authors themselves emphasize, further research is necessary, taking into account larger, more precise data. Nevertheless, this is a contribution to the diagnosis of elephant diseases, where a machine learning model was used. I recommend the manuscript for publication, with possible minor corrections.
Reviewer 3 Report
Comments and Suggestions for Authors
The submitted manuscript presents a retrospective analysis of 467 hospitalized Asian elephants at Thailand’s National Elephant Institute. Leveraging four routinely collected variables—age, sex, disease system, and length of stay (LOS)—the authors evaluate multiple classification models, identifying random forest as the best-performing algorithm. This is further complemented by survival models that estimate time to discharge or death across disease categories. Additionally, the authors provide a prototype tool developed in Excel/R to facilitate practical application.
This study offers potentially valuable insights to the existing body of literature on elephant health and clinical outcomes. However, prior to consideration for acceptance, the following comments should be addressed:
- Regarding the simple summary, it should smooth the prose and avoid sentence fragments; make the clinical utility claim precise (admission vs. in-hospital monitoring) once LOS usage is clarified.
- Survival analysis treats a composite event “time-to-discharge or death,” which conflates recovery with mortality. Standard practice would use competing-risks (cumulative incidence functions) or a multi-state model (Hospitalized → Discharged / Died). As written, Kaplan–Meier curves and Cox HRs on a composite endpoint are difficult to interpret clinically. Please re-analyze with cause-specific hazards or Fine–Gray, or model transitions explicitly.
- The classifier uses LOS to predict outcome. Although you state LOS is computed only up to the prediction/censoring time to avoid leakage, this still requires time that has already elapsed in hospital, so the model cannot support admission-time triage—your main stated need. Consider:
a) a day-0 model (no LOS),
b) landmarking models at fixed horizons (e.g., day 3, 7, 14), or
c) a joint/temporal model using accumulating daily features. Clearly separate these use-cases and report performance for at-admission predictions.
- Test-set performance for minority classes is modest (“class-wise recall 0.50, F1 0.54” before weighting), then improved after reweighting. It’s unclear whether class weighting and thresholds were tuned without peeking at the held-out test set. Please clarify the tuning protocol and report per-class precision/recall/F1 with 95% CIs for the final locked model, plus the confusion matrix. Avoid mixing cross-val metrics with post-hoc test adjustments.
- With only four predictors and single-center data, conclusions should be framed as proof-of-concept. Please temper language (e.g., “accurately stratify outcomes”) and emphasize the need for external validation before clinical deployment.
- Collapsing rare diagnoses into “Other” improves stability but harms interpretability and may mask high-risk entities. Therefore, the authors should provide a supplementary breakdown of “Other” by diagnosis and, if feasible, explore grouping schemes driven by clinical similarity rather than frequency alone.
- Table 1 excerpt appears truncated in places; double-check all tables for complete headers, units, and readable ranges.
- Finally, when stating hazard ratios (Table 5), the authors must specify reference groups (you do for sex and integument) and clearly state the event modeled once the survival framework is revised.
Comments on the Quality of English LanguageConcerning the quality of English, the manuscript presents numerous typos and grammar issues throughout. A careful language edit is needed. Moreover, they should unify terms (“disease system” vs. “disease group”; “eye problems” vs. “Eye”). Ensure consistent capitalization of algorithms and disease names.
Round 2
Reviewer 1 Report
Comments and Suggestions for Authors
The article has been thoroughly and sufficiently revised according to all comments
Reviewer 3 Report
Comments and Suggestions for Authors
The revised version submitted by the authors shows strong technical, methodological, and linguistic improvements that meet MDPI’s publication standards. Therefore, I recommend its publication in the present form